# Real-World Prevalence and Tolerability of Immune-Related Adverse Events in Older Adults with Non-Small Cell Lung Cancer: A Multi-Institutional Retrospective Study

**DOI:** 10.3390/cancers16112159

**Published:** 2024-06-06

**Authors:** Ryosuke Matsukane, Takahiro Oyama, Ryosuke Tatsuta, Sakiko Kimura, Kojiro Hata, Shuhei Urata, Hiroyuki Watanabe

**Affiliations:** 1Department of Pharmacy, Kyushu University Hospital, 3-1-1 Maidashi, Higashi-ku, Fukuoka 812-8582, Japan; 2Department of Pharmacy, Kumamoto University Hospital, 1-1-1 Honjo, Chuo-ku, Kumamoto 860-8556, Japan; pdeg2052@kuh.kumamoto-u.ac.jp; 3Department of Clinical Pharmacy, Oita University Hospital, 1-1 Idaigaoka, Hasamamachi, Yufu, Oita 879-5593, Japan; tatsuta@oita-u.ac.jp; 4Department of Pharmacy, Saga University Hospital, 5-1-1 Nabeshima, Saga 849-8501, Japan; mochinag@cc.saga-u.ac.jp; 5Department of Pharmacy, Fukuoka Tokushukai Hospital, 4-5 Sugukita, Kasuga-Shi, Fukuoka 816-0864, Japan; kojiro.hata@tokushukai.jp (K.H.); hiroyuki.watanabe@tokushukai.jp (H.W.); 6Department of Pharmacy, University of Miyazaki Hospital, 5200 Kiyotake-cho Kihara, Miyazaki 889-1692, Japan; s.urata@med.miyazaki-u.ac.jp

**Keywords:** immune checkpoint inhibitor, immune-related adverse events, older adults, non-small cell lung cancer

## Abstract

**Simple Summary:**

Immune checkpoint inhibitors (ICIs) represent a cornerstone in contemporary cancer therapy, yet managing immune-related adverse events (irAEs) remains pivotal. These events, characterized by reinvigorated autoimmune responses against normal tissues, present particular challenges, especially regarding their safety and tolerability in elderly patients. To address this gap, we conducted a multicenter retrospective cohort study focusing on patients with non-small cell lung cancer undergoing ICI therapy. Our findings revealed that irAE incidence, severity, and organ specificity did not significantly differ between elderly patients and their younger counterparts. However, elderly patients tended to transition to the best supportive care following irAE onset. These findings suggest that while age alone may not preclude ICI treatment, irAEs may be less tolerated in certain elderly individuals, potentially impacting patient prognosis. Identifying markers of irAE intolerance, such as frailty, sarcopenia, and cachexia, alongside chronological age, could aid in optimizing patient selection and clinical benefits of ICI treatment in this population.

**Abstract:**

With cancer diagnosis occurring at older ages, the use of immune checkpoint inhibitors (ICIs) has extended to older adults. However, the safety of immune-related adverse events (irAEs) in this population remains unclear and relies on data extrapolated from younger adults. This multicenter retrospective study aimed to examine irAE prevalence and tolerability in older adults. We included 436 patients with non-small lung cancer undergoing ICI therapy and dichotomized them into two age groups (< or ≥75 years). Incidence of any irAE grade, grade ≥3 irAEs, and steroid usage after irAE occurrence was similar between younger (n = 332) and older groups (n = 104). While the younger patients with irAEs showed prolonged overall survival in the 12-month landmark Kaplan–Meier analysis (Hazard ratio (HR) 0.59, 95% confidence interval (CI) 0.38–0.89, *p* = 0.013), the older cohort did not (HR 0.80, 95% CI 0.36–1.78, *p* = 0.588). Although no differences were observed with ICI continuation or re-challenge after irAE onset, the elderly cohort had double the irAE cases that required a transition to best supportive care (BSC) (11.3% vs. 22.4%, *p* = 0.026). In conclusion, although irAE prevalence remains consistent regardless of age, the increased conversion to BSC post-irAE onset in older adults suggests diminished tolerability and the potential absence of favorable prognosis associated with irAEs in this population.

## 1. Introduction

Non-small cell lung cancer (NSCLC) is the leading cause of cancer-related fatalities globally and accounts for the highest mortality rate among both men and women [1]. In the USA, the median age at which NSCLC is diagnosed is 71 years old, according to the Surveillance, Epidemiology, and End Results (SEER) 22 cancer registry [2]. In addition, 36.4% of cases are diagnosed at >75 years of age, which has escalated to 58.8% in the Japanese population [2,3]. Given these statistics, there is an urgent need to evaluate the efficacy and safety of cancer treatments in older adults. However, the current evidence for their management is largely extrapolated from existing literature focused on younger adults.

The elderly population is considerably underrepresented in clinical trials that establish standards for the efficacy and safety of cancer treatments. Fewer than 10% of patients aged >70 years participate in National Cancer Institute-sponsored clinical trials [4]. Furthermore, the age disparities between clinical trials and real-world treatment populations are larger in lung cancer than in other tumors [5]; thus, post-marketing investigations are crucial for establishing the efficacy and safety profile in elderly patients.

Immune checkpoint inhibitors (ICIs) have emerged as indispensable frontline therapies to manage advanced or recurrent NSCLCs without druggable mutations [4]. However, considering the age-associated immunosenescence, confirming ICI efficacy in elderly patients is necessary, and several studies have provided valuable insights [5,6,7,8]. Moreover, ICIs cause a unique array of adverse effects, termed immune-related adverse events (irAEs), wherein reinvigorated immune responses may inadvertently target healthy organs [9]. Given the potential severity of irAEs, establishing effective management strategies, including safety and tolerability in elderly patients, is of paramount importance.

Several studies have investigated the occurrence of irAEs in the elderly population. An integrated analysis encompassing three clinical trials that evaluated pembrolizumab demonstrated that irAE incidence in elderly patients with NSCLC was comparable to that in younger cohorts [5]. Morinaga et al. conducted a real-world clinical study and observed that the irAE profile remained consistent across age groups [7]. However, Tsukita et al. reported that combining ICI with chemotherapy in patients aged ≥75 years did not confer survival benefits and was associated with an increased incidence of grade ≥3 irAEs compared to ICI monotherapy [8]. Thus, information on irAEs in the elderly population remains unclear, and accumulating evidence in real-world clinical settings is necessary.

To assess the safety of irAEs, understanding post-irAE outcomes and their prevalence is crucial. While prior studies have highlighted irAEs as favorable prognostic indicators in ICI therapy [10], effective management and mitigation of irAEs remain imperative to achieve sustained ICI responses. Some irAEs can result in significant organ toxicity, which often necessitates aggressive treatment with high-dose steroids or immunosuppressive drugs. The elderly population is more susceptible to being burdened with such treatments after an irAE occurs. Most previous studies examining irAEs in elderly patients have primarily focused on their prevalence rather than assessing the true tolerability and subsequent clinical outcomes of these adverse events. This gap in the literature leaves unanswered questions about whether elderly patients can effectively tolerate ICI therapy and derive benefits after the occurrence of irAEs. Our study aimed to address this gap by comprehensively evaluating not only the incidence but also the management and outcomes of irAEs in elderly patients.

Consequently, we conducted a multicenter, retrospective cohort study to investigate patients with NSCLC undergoing ICI therapy in real-world clinical settings. Our objective was to delineate the safety and tolerability of ICI therapy among elderly patients by scrutinizing irAE incidence and subsequent patient outcomes, including the ability to overcome irAEs.

## 2. Materials and Methods

### 2.1. Study Design, Patients, and Data Collection

This retrospective, multi-institution, observational cohort study conformed to the STROBE (cohort study) guidelines [11]. The inclusion criteria for this study comprised patients with advanced or metastatic NSCLC who underwent ICI therapy between September 2014 and March 2022 at Kyushu University Hospital and between April 2020 and March 2022 at Saga University Hospital, Kumamoto University Hospital, University of Miyazaki Hospital, Oita University Hospital, and Fukuoka Tokushukai Hospital. The ICIs administered included nivolumab, pembrolizumab, atezolizumab, and ipilimumab. The exclusion criteria were patients lost to follow-up or those who received ICIs during clinical trials. The follow-up period concluded on 31 October 2023. According to the Japanese Lung Cancer Society Guidelines for NSCLC, patients aged ≥75 years were defined as elderly. Patient data collected from the medical records of each hospital at the initiation of treatment included age, sex, Eastern Cooperative Oncology Group Performance Status (ECOG PS), disease stage, tumor histology, metastasis, programmed cell death ligand-1 (PD-L1) tumor proportion score (TPS), treatment line, and administered ICI regimen. After treatment initiation, we collected the treatment outcome data (date of disease progression, date of death, and last confirmed survival date) and irAE information, such as date of onset, severity grade, usage and amount of steroid or immunosuppressant agents, and the actions taken after irAE onset (e.g., continuous or re-challenge ICIs, watchful waiting, switch to subsequent treatment, and switch to best supportive care). IrAEs were classified according to established guidelines [12,13,14], and their severity was determined using the National Cancer Institute Common Terminology Criteria for Adverse Events v.5.0. ICI re-challenge was defined as resuming ICI treatment after irAE development, irrespective of a withdrawal period. The irAEs were categorized based on peak steroid dose as follows: no administration, low (<0.5 mg/kg prednisolone (PSL) equivalent), high (0.5–2.0 mg/kg PSL equivalent), and intravenous methylprednisolone (mPSL) pulse therapy. 

### 2.2. Ethical Statements

This study was approved by the Institutional Review Boards of Kyushu University Graduate School and Faculty of Medicine (Approval No. 22232-00), Saga University Hospital (Approval No. 2022-C-63), Kumamoto University Hospital (Approval No. 2686), University of Miyazaki Hospital (Approval No. O-1286), Oita University Hospital (Approval No. 2453-C65), and Fukuoka Tokushukai Hospital (Approval date 21 January 2023). This study was conducted in accordance with the principles of the Declaration of Helsinki. The requirement for informed consent was waived owing to the retrospective nature of the study, and patients were given the option to opt out of the study via our official website.

### 2.3. Statistical Analysis

Competing risk analysis was used to calculate the cumulative incidence of irAEs. Statistical analyses were performed using Gray’s test with a Bonferroni correction, and death without irAEs was defined as a competing risk. Survival probabilities were analyzed using the Kaplan–Meier method, and differences between groups were compared using the log-rank test with a Bonferroni correction. We utilized a logistic regression model incorporating sex, performance status (PS), and treatment line and administered ICIs as covariates for propensity score matching. Given that differences in the treatment duration of ICIs can influence the prevalence of irAEs, we selected covariates that impact prognosis and varied between the younger and older cohorts. The patients were divided into one-to-one groups using nearest-neighbor matching with a 0.2 caliper width. Multivariate analysis was performed using logistic regression analysis. Data from the two groups were compared using the Mann–Whitney U test and unpaired *t*-test. Fisher’s exact test was used to compare clinical variables in patients dichotomized by age. All statistical analyses were performed using GraphPad Prism version 9.4.0 (GraphPad Software, La Jolla, CA, USA); EZR version 1.55 (Saitama Medical Center, Jichi Medical University, Saitama, Japan) [15], which is a graphical user interface for R (The R Foundation for Statistical Computing, Vienna, Austria); and JMP version 16.0.0 (SAS Institute, Inc., Cary, NC, USA). All tests were two-sided, and *p*-values < 0.05 were considered statistically significant.

## 3. Results

### 3.1. Patient Selection and Characteristics

We collected medical records of 436 patients with advanced and metastatic NSCLC who received ICI treatment at six hospitals in Japan. Table 1 shows the clinical backgrounds and characteristics of patients stratified by age group (<75 years and ≥75 years). Among the 436 patients, 104 (23.9%) fell in the elderly category. Most of the baseline characteristics were comparable between the younger and older cohorts. However, a higher proportion of patients in the elderly group received ICI monotherapy (<75 years: 49.4%, ≥75 years: 73.1%), while significantly fewer were treated with a chemotherapy combination (<75 years: 35.8%, ≥75 years: 21.2%) and especially programmed cell death protein 1 (PD-1) + cytotoxic T-lymphocyte associated protein 4 (CTLA-4) therapy (<75 years: 14.8%, ≥75 years: 5.8%).

### 3.2. Cumulative Incidence of irAEs in Younger and Older Patients

In our analysis comparing younger (n = 332) and older groups (n = 104), we observed similar median progression-free survival (PFS) times (<75 years: 5.6 months vs. ≥75 years: 5.0 months, *p* = 0.139, Figure 1a). Throughout the observation period, we documented 275 cases of irAEs in 181 younger patients (average of 1.52 irAEs per patient) and 67 cases in 50 elderly patients (average of 1.34 irAEs per patient). The cumulative incidence of irAEs of any grade was comparable between the two groups, with 38.6% and 33.2% at three months and 51.1% and 45.6% at 12 months in the younger and elderly groups, respectively (Figure 1b). Likewise, the onset of grade ≥3 irAEs was consistent between the groups, with 10.9% and 9.7% at three months and 15.8% and 15.9% at 12 months in the younger and elderly groups, respectively (Figure 1c). Although skin toxicity was less prevalent in the elderly patients than in their younger counterparts, other irAEs displayed similar occurrence rates in both age groups (Figure 1d). In addition, no significant differences were found in the distribution of severe-grade irAEs across various organs between the two groups (Figure 1e). These results were consistent when compared across each regimen (Appendix A).

### 3.3. Cumulative Incidence of irAEs in Propensity Score-Matched Patients

Owing to disparities in treatment regimens between age-stratified groups, particularly the higher usage of anti-PD-1 and CTLA-4 inhibitor combination therapy in younger patients, we compared irAE onset in patients with equivalent backgrounds. Using propensity score matching, we selected 103 pairs of patients and adjusted them for sex, PS, and treatment line, and we administered ICIs as covariates. The results demonstrated similar background characteristics (Appendix A), with both groups exhibiting a similar median PFS (<75 years: 5.3 months vs. ≥75 years: 5.0 months, *p* = 0.359, Appendix A). The cumulative incidence of irAEs of any grade was also akin between the two groups, with 31.4% and 33.5% at three months and 45.8% and 46.0% at 12 months in the younger and elderly groups, respectively (Appendix A). However, the onset of grade ≥3 irAEs was slightly higher in elderly patients, with 8.8% and 9.8% at three months and 9.8% and 16.0% at 12 months (Appendix A). No significant differences were observed in irAE distribution across various organs between the two age groups (Appendix A).

### 3.4. Steroid Treatment Post-irAE Occurrence

Among the 332 younger patients, 181 (54.5%) developed irAEs, and 90 (27.1%) subsequently received corticosteroid therapy. This therapy was administered in varying doses: 45 patients (13.6%) received low doses (<0.5 mg/kg PSL equivalent), 28 (8.4%) received high doses (≥0.5 mg/kg PSL equivalent), and 17 (5.1%) received mPSL pulse therapy. In contrast, 50 (48.1%) of the 104 elderly patients developed irAEs, of whom 23 (22.1%) received corticosteroid therapy. The distribution of doses administered was as follows: 9 patients (8.7%) received low doses and 11 (10.6%) received high doses, while three (2.9%) received mPSL pulse therapy. Table 2 demonstrates no significant differences in the usage and dosage of steroids for irAEs that developed across various organs between the two groups.

### 3.5. irAE Incidence as a Prognostic Marker

Acknowledging the established prognostic benefits of irAEs, our study investigated their impact on patient survival by examining both younger and older cohorts. Among younger patients, the presence of irAEs correlated with significant extensions in both PFS (hazard ratio (HR) 0.45, 95% confidence interval (CI) 0.35–0.58, *p* < 0.0001, Figure 2a) and overall survival (OS) (HR 0.49, 95% CI 0.37–0.65, *p* < 0.0001, Figure 2b) compared with those without irAEs (Appendix A). To avoid lead-time bias caused by the time-dependent nature of irAE occurrence, we conducted a 12-month landmark survival analysis, and the findings remained consistent (HR 0.59, 95% CI 0.38–0.89, *p =* 0.013, Figure 2c). Conversely, among patients aged 75 years or older, those experiencing irAEs exhibited prolonged PFS (HR 0.33, 95% CI 0.21–0.52, *p* < 0.0001, Figure 2d) and OS (HR 0.39, 95% CI 0.24–0.64, *p* = 0.0002, Figure 2e) relative to those without irAEs (Appendix A). However, in the 12-month landmark analysis, irAEs did not confer a prognostic advantage as observed in the younger cohort (HR 0.80, 95% CI 0.36–1.78, *p* = 0.588, Figure 2f).

### 3.6. Clinical Outcome after irAE Onset

We categorized the developed irAEs based on subsequent management strategies as follows: continuation or re-challenge of ICIs, discontinuation of ICIs followed by switching to subsequent treatment, discontinuation of ICIs with a period of watchful waiting, and discontinuation of ICIs with transition to best supportive care (BSC). No disparities were observed in cases where ICIs were continued or re-challenged after irAE onset, switched to subsequent treatments, or underwent a period of watchful waiting (Figure 3a). However, the proportion of irAE cases requiring a transition to BSC without further anti-cancer treatment nearly doubled in the elderly cohort (<75 years: 11.3% vs. ≥75 years: 22.4%, *p* = 0.026, Figure 3a). No significant differences were revealed in irAE severity (Figure 3b) or steroid usage (Figure 3c) of continuing or re-challenging ICI treatment between the two groups. Among all irAEs, skin toxicities exhibited a higher rate of treatment restart (Figure 3d). In contrast, elderly patients who experienced grade ≥3 irAEs demonstrated higher BSC transition rates than younger patients (<75 years: 22.6% vs. ≥75 years: 47.4%, *p* = 0.046, Figure 3e). Although patients who received high doses of steroids following irAE onset showed a similar rate of BSC transition in both groups, elderly patients who received no or low doses of steroids showed significantly higher rates of BSC transition than those in the younger cohort (<75 years: 6.7% vs. ≥75 years: 20.8%, *p* = 0.004, Figure 3f). Interstitial pneumonitis emerged as the most common irAE, leading to BSC conversion regardless of the age (Figure 3g). Finally, we conducted a multivariate analysis to identify risk factors for patients transitioning to BSC after irAE onset (n = 41). The analysis revealed that age ≥75 years (odds ratio 2.47, 95% CI 1.11–5.49, *p* = 0.027) and ECOG PS ≥2 (odds ratio 3.17, 95% CI 1.05–9.56, *p* = 0.041) were independent risk factors (Table 3).

## 4. Discussion

This multicenter retrospective study was conducted to comprehensively investigate irAEs and assess the safety and tolerability of ICI treatment in older adults. While prior research has analyzed irAE onset and severity based on patient age, our study uniquely contributes information regarding how the elderly cohort manages irAEs and whether they overcome irAE onset in real-world settings compared with younger patients. Our findings suggest no disparities in the incidence or severity of irAEs between age groups, and these observations remained consistent even after adjusting for patient characteristics. The higher incidence of skin-related irAE in the younger group may be due to the greater use of anti-PD-1 plus anti-CTLA-4 combination therapy. After adjusting for patient background, including treatment regimens, with propensity score matching, those incidences were comparable between the two groups.

Although the emergence of irAEs has been associated with favorable prognostic outcomes [10,16,17,18], acknowledging that severe irAEs can precipitate treatment discontinuation and potentially threaten patient lives is crucial. Our analysis revealed that younger patients who experienced irAEs had extended PFS, OS, and 12-month landmark OS. In contrast, elderly patients did not experience a similar prognostic advantage in 12-month landmark OS. Examination of post-irAE management strategies indicated similar rates of continuation or re-challenge of ICI treatment, switching to subsequent therapies, and watchful waiting between the two age groups. However, elderly patients exhibited a significantly higher propensity for BSC transition following irAE onset. These findings suggest that some irAEs may be less tolerated in elderly populations than in their younger counterparts, and that irAEs may not confer prognostic benefits in older patient cohorts.

Elderly patients are often underrepresented in clinical trials, which leads to extrapolation of treatment efficacy and safety data from younger cohorts. However, integrated analyses of previous clinical trials have demonstrated a significantly prolonged prognosis in elderly patients with untreated NSCLC receiving ICI therapy compared to those receiving chemotherapy alone, mirroring the outcomes seen in younger patients [5]. Moreover, real-world evidence suggests a comparable treatment efficacy between younger and older populations [6,7,19]. Our results also showed that chronological age did not affect PFS after ICI treatment, which is consistent with these results. Consequently, as recommended by the international expert panel, age *per se* is not a limitation for ICI treatment in terms of efficacy [20].

Several studies have evaluated the safety of ICI treatment in elderly patients. Nebhan et al. examined the safety of ICI monotherapy in patients older than 80 years and reported 41.3% and 12.2% incidences of irAEs and severe-grade irAEs, respectively [21]. In addition, clinical data from France and Canada have revealed that the prevalence and organ specificity of irAEs in NSCLC do not differ with age [22,23]. The results of these studies are generally consistent with our results in a Japanese population. In contrast, an extensive analysis using the Food and Drug Administration Adverse Event Reporting System (FAERS) database reported a significant increase in the number of reported irAEs in an elderly cohort, especially in cardiovascular and pulmonary disorders [24]. Pharmacovigilance studies face certain limitations, such as reporting bias and difficulties in differentiating between irAEs and other treatment-related adverse events. However, information on rare irAEs such as myocarditis should be taken seriously because collecting and analyzing these events in real-world clinical settings can be challenging [25].

The incidence of irAEs is a favorable prognostic factor for ICI therapy; however, to achieve this benefit, patients need to overcome its toxicity and receive subsequent immunosuppressive treatment. Therefore, the true tolerability of irAEs should be evaluated based on the clinical outcomes after irAE development, along with their prevalence and organ specificity. In this study, while no significant difference was observed in the incidence and severity of irAEs between the younger and older cohorts, a notable disparity emerged in the number of irAE cases necessitating transition to BSC, particularly in the elderly cohort. This increase in BSC transitions among older patients can be attributed to two main factors: a higher rate of BSC transition in grade ≥3 irAEs and an increased rate of transition in mild irAEs not requiring high-dose steroids. These findings highlight a distinct vulnerability among older patients to the toxicity of irAEs. Possible explanations include their reduced capacity to fully recover from irAE toxicity or prolonged recovery periods that compromise their performance status, thereby hindering their ability to pursue further treatment and allowing disease progression. These insights emphasize the unique challenges faced by elderly patients undergoing ICI therapy and underscore the necessity of tailored management strategies. Furthermore, our findings indicate that despite the development of irAEs, the 12-month landmark OS was not prolonged in elderly patients. This suggests the presence of an ICI-unfit population among the elderly cohort, who may struggle to overcome irAEs and derive ICI benefits. These results underscore the importance of identifying and addressing the specific needs of elderly patients undergoing ICI therapy. 

This study had some limitations. Although we compared the relationship between irAEs and chronological age in this multicenter study, the number of individual irAEs was small, and rare irAEs could not be analyzed. The number of patients treated with each regimen, such as anti-PD-1 combined with anti-CTLA-4 antibody therapy, was small. It is important to note that the incidence of grade ≥3 irAEs in this regimen was nearly 40% [26]. In addition, these combinations may induce uncommon irAEs such as myocarditis and cytokine release syndrome [27,28,29]. We anticipate that the proportion of patients who will not tolerate these treatments and will be compelled to switch to BSC will increase if these regimens are administered to the elderly. Further analysis with a larger population is required to fully understand the irAE tolerability of these regimens. In this study, patients aged ≥75 years were defined as elderly according to the guidelines of the Japanese Lung Cancer Association; however, a recent expert panel evaluating immunotherapy in the older population mentioned that elderly patients should be considered only as a surrogate for clinical factors of frailty [20]. Cognitive function and life-space mobility, but not age, have been reported as risk factors for irAEs in older adults [30]. As this study was a retrospective analysis, we could not examine other surrogate markers in elderly patients, including comorbidities. Further research is required to adequately identify frailty among the elderly population.

## 5. Conclusions

In summary, our findings indicated that the incidence and severity of irAEs, along with their organ specificity, do not significantly differ between elderly patients with NSCLC and their younger counterparts. However, elderly patients demonstrated a higher conversion rate to BSC following irAE onset. These results suggest that while age itself may not preclude ICI treatment, irAEs may be less tolerated in certain elderly individuals and may not confer a favorable prognostic factor in this population. It is imperative to incorporate additional surrogate markers of frailty, such as sarcopenia and cachexia, along with chronological age, to accurately identify individuals who can tolerate irAEs and derive clinical benefits from ICI therapy.

## Figures and Tables

**Figure 1 cancers-16-02159-f001:**
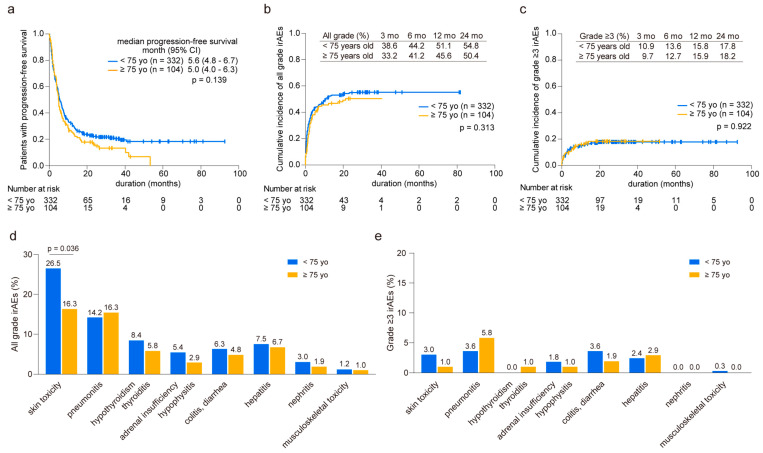
Prevalence of immune-related adverse events in younger and older patients. (**a**) Kaplan–Meier curve analysis depicting progression-free survival in younger (<75 years old, n = 332) and older (≥75 years old, n = 104) patients. (**b**) Cumulative incidence of all-grade and (**c**) grade ≥3 immune-related adverse events (irAEs). The prevalence of irAEs is calculated at 3, 6, 12, and 24 months. (**d**) Comparison of organ specificity in all grades and (**e**) grade ≥3 irAEs between younger and older patients. Significance was determined using the (**a**) log-rank test, (**b**,**c**) Gray’s test, and (**d**,**e**) Fisher’s exact test.

**Figure 2 cancers-16-02159-f002:**
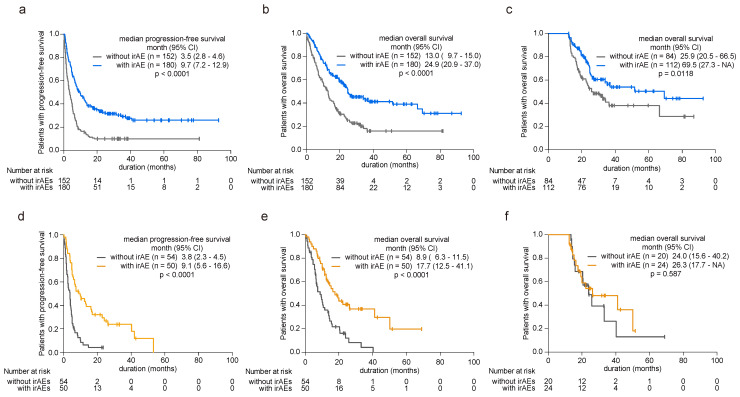
Influence of irAE onset on patient prognosis in younger and older patients. (**a**) Kaplan–Meier curve analysis depicting progression-free survival (PFS) and (**b**) overall survival (OS) in younger patients (<75 years old) with or without irAE development. (**c**) The 12-month landmark Kaplan–Meier analysis assessing OS in younger patients with or without irAE development. (**d**) Kaplan–Meier curve analysis for PFS and (**e**) OS in elderly patients (≥75 years old) with or without irAE development. (**f**) The 12-month landmark Kaplan–Meier analysis assessing OS in elderly patients with or without irAE development. Significance was determined using the log-rank test.

**Figure 3 cancers-16-02159-f003:**
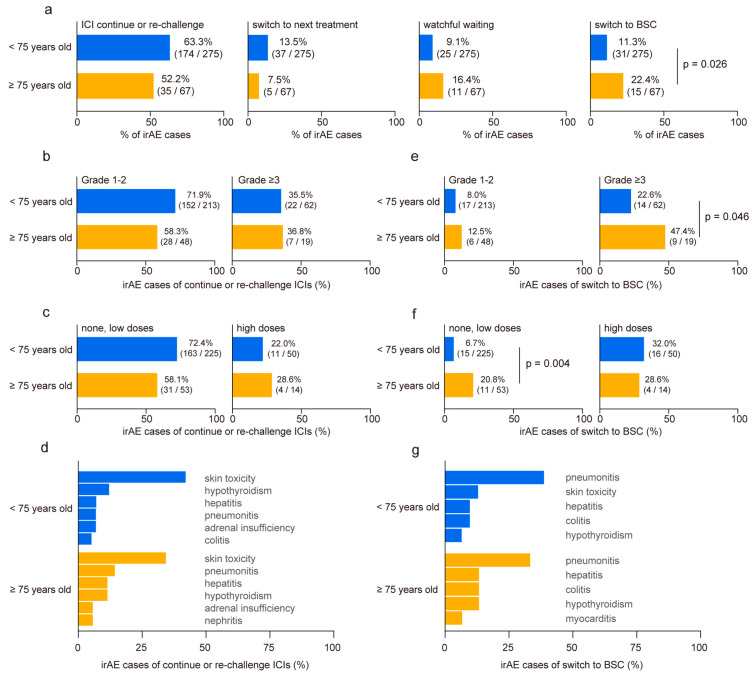
Comparison of clinical outcome post-immune-related adverse event occurrence in younger and older patients. (**a**) Comparison of clinical outcomes after immune-related adverse events (irAEs) in younger (<75 years old) and older (≥75 years old) patients. Post-irAE management is classified as follows: continuation or re-challenge of immune checkpoint inhibitors (ICIs), discontinuation of ICIs followed by switching to a subsequent treatment, discontinuation of ICIs with a period of watchful waiting, and discontinuation of ICIs with the transition to best supportive care (BSC). (**b**) Relationship between age group and irAE severity and (**c**) steroid doses in cases where ICIs were continued or re-challenged after irAE onset. (**d**) Ratio of irAE toxicity in cases that continued or re-challenged ICI treatment. (**e**) Relationship between age group and irAE severity and (**f**) steroid doses in irAE cases necessitating transition to BSC without further anti-cancer treatment. (**g**) Ratio of irAE toxicity leading to BSC conversion. Significance was determined using Fisher’s exact test.

**Table 1 cancers-16-02159-t001:** Patient cohort characteristics.

Characteristics	<75 Years Age		≥75 Years Age		*p*
	(n = 332)		(n = 104)		
Age, median—years (range)	66	(36–74)	78	(75–89)	<0.001
Sex—no. (%)					0.405
Male	260	(78.3)	86	(82.7)	
Female	72	(21.7)	18	(17.3)	
ECOG PS—no. (%)					0.999
0–1	291	(87.7)	91	(87.5)	
≥2	41	(12.3)	13	(12.5)	
Histology—no. (%)					0.303
Adenocarcinoma	209	(63.0)	61	(58.7)	
Squamous	84	(25.3)	34	(32.7)	
Other	39	(11.7)	9	(8.7)	
PD-L1 status—no. (%)					0.577
TPS ≥50%	111	(33.4)	37	(35.6)	
TPS 1–49%	101	(30.4)	37	(35.6)	
TPS <1%	67	(20.2)	16	(15.4)	
not investigated	53	(16.0)	14	(13.5)	
Common sites of metastasis—no. (%)					
brain	88	(26.5)	22	(21.2)	0.302
bone	103	(31.0)	32	(30.8)	0.999
liver	32	(9.6)	7	(6.7)	0.435
Treatment line—no. (%)					0.724
1st line	198	(59.6)	63	(60.6)	
2nd line	72	(21.7)	25	(24.0)	
3rd line or more	62	(18.7)	16	(15.4)	
Administrated ICIs—no. (%)					<0.001
Anti-PD-1/PD-L1 monotherapy	164	(49.4)	76	(73.1)	
nivolumab	76	(22.9)	20	(19.2)	
pembrolizumab	76	(22.9)	46	(44.2)	
atezolizumab	12	(3.6)	10	(9.6)	
Anti-PD-1/PD-L1 with chemotherapy	119	(35.8)	22	(21.2)	
pembrolizumab + chemotherapy	93	(28.0)	17	(16.3)	
atezolizumab + chemotherapy	26	(7.8)	5	(4.8)	
Anti-PD-1 + anti-CTLA-4 combination	49	(14.8)	6	(5.8)	
nivolumab + ipilimumab	10	(3.0)	3	(2.9)	
nivolumab + ipilimumab + chemotherapy	39	(11.7)	3	(2.9)	

ECOG PS, Eastern Cooperative Oncology Group Performance Status; PD-L1, programmed cell death ligand-1; TPS, tumor proportion score; ICI, immune checkpoint inhibitor; PD-1, programmed cell death protein-1; CTLA-4, cytotoxic T-lymphocyte associated protein-4.

**Table 2 cancers-16-02159-t002:** Comparison of steroid treatment after irAE in younger and older patients.

Characteristic	n	Steroid Usage					
		Any Dose		*p*	High Dose		*p*
Skin toxicity							
<75 years old	88	20	(22.7)	0.759	7	(8.0)	>0.9999
≥75 years old	17	3	(17.6)		1	(5.9)	
Pneumonitis							
<75 years old	47	31	(66.0)	>0.9999	20	(42.6)	0.7818
≥75 years old	17	11	(64.7)		8	(47.1)	
Hypothyroidism, thyroiditis							
<75 years old	28	0	(0.0)	>0.9999	0	(0.0)	>0.9999
≥75 years old	6	0	(0.0)		0	(0.0)	
Adrenal insufficiency							
<75 years old	18	17	(94.4)	>0.9999	3	(16.7)	>0.9999
≥75 years old	3	3	(100.0)		0	(0.0)	
Colitis, diarrhea							
<75 years old	21	12	(57.1)	>0.9999	7	(33.3)	>0.9999
≥75 years old	5	3	(60.0)		1	(20.0)	
Hepatitis							
<75 years old	25	4	(16.0)	0.5896	4	(16.0)	>0.9999
≥75 years old	7	2	(28.6)		1	(14.3)	
Nephritis							
<75 years old	10	2	(20.0)	>0.9999	1	(10.0)	>0.9999
≥75 years old	2	0	(0.0)		0	(0.0)	
Musculoskeletal toxicity							
<75 years old	4	2	(50.0)	>0.9999	0	(0.0)	>0.9999
≥75 years old	1	0	(0.0)		0	(0.0)	

irAE, immune-related adverse events.

**Table 3 cancers-16-02159-t003:** Multivariate logistic regression analysis of patients transitioned to BSC after irAE.

Characteristic	Odds Ratio	95% CI	*p*
Age			
<75 years old	ref		
≥75 years old	2.47	1.11–5.49	0.027
Sex			
Male	ref		
Female	1.40	0.55–3.62	0.478
ECOG PS			
0–1	ref		
≥2	3.17	1.05–9.56	0.041
Treatment line			
1st line	ref		
2nd line or more	2.10	0.90–4.86	0.084
Administrated ICIs—no. (%)			
Anti-PD-1/PD-L1 monotherapy	ref		
Anti-PD-1/PD-L1 with chemotherapy	0.81	0.32–2.06	0.663
Anti-PD-1 + anti-CTLA-4 combination	1.07	0.34–3.34	0.909

95% CI, 95% confidence interval; ECOG PS, Eastern Cooperative Oncology Group Performance Status; ICI, immune checkpoint inhibitor; PD-1, programmed cell death protein-1; PD-L1, programmed cell death ligand-1; CTLA-4, cytotoxic T-lymphocyte associated protein-4.

## Data Availability

The data can be shared up on request.

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
