# Peer review of "Real-World Prevalence and Tolerability of Immune-Related Adverse Events in Older Adults with Non-Small Cell Lung Cancer: A Multi-Institutional Retrospective Study"

_cancers, 2024, doi:10.3390/cancers16112159_

Round 1

Reviewer 1 Report

Comments and Suggestions for Authors

I am very pleased to be able to review your manuscript. While the study is worthy of acceptance, several concerns should be addressed, which are discussed below.

1. the introduction effectively outlines the significance of immune checkpoint inhibitors (ICIs) and associated immune-related adverse events (irAEs) in elderly non-small cell lung cancer patients. However, this section would benefit from a more detailed discussion of the gaps in the existing literature and how this study aims to address them.

2. a multicenter retrospective design is appropriate for the purpose of the study. However, additional details regarding patient selection criteria and potential bias in patient selection would increase the rigor of the study.

3. the patient characteristics table is comprehensive. For greater clarity, consider including a column of p-values comparing baseline characteristics between younger and older cohorts to more clearly highlight significant differences.

4. the use of Kaplan-Meier curves and competing risk analysis is commendable. However, the methodology section should explain the process of propensity score matching in more detail, including the covariates used and the rationale for their selection.

5. Although the results are clearly presented, the discussion could benefit from a deeper interpretation of the findings. For example, explore potential reasons why older patients have a higher conversion rate to best supportive care after the onset of irAE.

Comments on the Quality of English Language

Minor editing of English language required.

Reviewer 2 Report

Comments and Suggestions for Authors

The authors examined and reported adverse events in elderly and non-elderly patients receiving ICI treatment, using 75 as the cutoff. They appropriately cited relevant literature and summarized their findings. However, I have one question.

Please refer to the following comment:

  1. In the non-elderly group, regimens containing anti-CTLA-4 antibodies are frequently used. Recently, there have been reports of increased adverse events with regimens that include anti-CTLA-4 antibodies (Ann Oncol. 2023 Nov;34(11):1064-1065.). Given that regimens containing Ipilimumab are frequently administered in the non-elderly group, could the differences in regimens affect the conclusions of the authors' paper? Would additional analysis or mention in the Discussion section be necessary to address this impact?

Reviewer 3 Report

Comments and Suggestions for Authors

This study examines the real-world prevalence and tolerability of irAEs in older adults with NSCLC undergoing ICI therapy. It finds that while the incidence and severity of irAEs are similar between older and younger patients, older adults are more likely to transition to best supportive care following irAE onset, indicating a lower tolerability. The study suggests additional markers like frailty and cachexia should be considered to optimize patient selection and outcomes.

The categorization of irAEs and their severity according to established guidelines. I would suggest the authors consider more potential confounders, such as comorbidities and prior treatment history to provide a more meaningful guidance on the patient selection.

The incidence of irAEs was comparable between age groups, with no significant differences in organ-specific irAEs or severe grade ≥3 irAEs. These findings are consistent with prior studies. However, the study could benefit from a deeper exploration of why certain irAEs, like skin toxicity, were less prevalent in the elderly. 

Round 2

Reviewer 1 Report

Comments and Suggestions for Authors

The authors appropriately responded to my concerns. Thank you.